# Dynamic Evaluation of Early Silvicultural Treatments for Wildfire Prevention

**Míriam Piqué [1],\*, José Ramón González-Olabarria [1]**  **and Eduard Busquets [2]**

1   Joint Research Unit CTFC-AGROTECNIO, Carretera de Sant Llorenç de Morunys, Km 2, 25280 Solsona, Spain; jr.gonzalez@ctfc.cat
2   Forest Science and Technology Centre of Catalonia (CTFC), Carretera de Sant Llorenç de Morunys, Km 2, 25280 Solsona, Spain; eduard.busquets@ctfc.cat
\*   Correspondence: miriam.pique@ctfc.cat

**Abstract:** Thinning young forest stands is a common practice to improve the future development of the remaining trees and enhance their resistance to abiotic and biotic disturbances. The objective of this study was to consider the effectiveness of precommercial thinning, over time, implemented on *Pinus halepensis* (Aleppo pine) thickets, regarding fuel evolution and potential fire behavior. For this purpose, we established 44 plots on untreated and thinned Aleppo pine stands, measured all of the relevant fuel characteristics and simulated fire behavior under average and extreme fire weather scenarios. The plots were at different stages of fuel evolution (0.5 to 10 years since treatment, plus untreated stands), so that the evolution of the variables defining forest structure and the amount and distribution of surface fuels could be captured. The results show that precommercial thinning, when accompanied with pruning and surface fuel management, had a clear impact on fire behavior and on the potential of fire crowning during the first two to four years after the treatment. After that initial period, the buildup of understory vegetation minimized treatment effectiveness in mitigating potential fire behavior. In general, it can be stated that precommercial thinning has a positive impact on fire mitigation, but the impact that opening the tree canopy has on ground vegetation development must be considered in order to plan more efficient management strategies.

**Keywords:** precommercial thinning; preventive silviculture; fuel management; fire simulation; Nexus; fire behavior; crown fire; *Pinus halepensis*



## 1. Introduction

Silvicultural practices are planned and applied to retain or improve different forest services. The influence on the yield of those services over time depends on the sudden impact that this type of managerial activities has on the present state of a forest, and the influence that such changes will have on the future evolution of the forest. One of the aspects that should be considered when anticipating the impact of silvicultural actions is their impact on modifying the risk that disturbances entail for forests and their associated functions [1]. In the case of wildland fires, it is widely understood that any managerial activity that modifies the amount and distribution of living or dead fuels will have a direct impact on fire behavior, severity and expected damage [2,3]. In principle, by decreasing the amount of living or dead surface fuels, the spread and intensity of surface fires can be limited [4,5], and by creating a discontinuity between surface fuels and the lowest branches of trees, the probability of fire crowning can be also reduced [6].

Among the silvicultural practices often recommended to enhance the forests' resistance to fire, applying early precommercial thinning to thickets offers some of the more tangible benefits. By reducing crown density, thinning decreases the potential transmission of fire between neighboring trees [7]. Furthermore, the increased radial growth of the remaining individuals will produce larger trees earlier [8], which also will translate into thicker bark and increased resistance of individual trees to fire [9]. However, all of those positive impacts

of thinning on increasing fire resistance depend largely on complementing thinning with ground treatments aimed at reducing the generated slash [2].

Across the Mediterranean basin, *Pinus halepensis* Mill. (Aleppo pine) constitutes one of the main components of the forests at lower altitudes [10]. Aleppo pine is a highly flammable species whose post-fire regeneration strategy consists of releasing large amounts of seeds from serotinous cones [11,12]. These characteristics help to explain the great colonization potential of this species after intense fires, and at the same time the proclivity of crown fire occurrence in Aleppo pine forests. In highly populated coastal regions in the northern Mediterranean Basin, such as Catalonia, where Aleppo pine forests cover more than 130,000 hectares [13], this tendency of high intensity fires, accompanied by the higher rates of tree mortality due to fires in the region [14], constitutes a challenge to be considered when planning fire and forest management strategies. Out of the management strategies considered to diminish the probability of hazardous crown fires, precommercial thinning of highly dense young stands is one of the most commonly used, due to its combined advantages in terms of fire hazard reduction, tree growth and drought resistance enhancement [15–17]. Still, once the evolution of fuels is considered, there is limited knowledge on the effectiveness of this type of practices over time. Following this idea, Palmero-Iniesta et al. [16] evaluated not only the immediate impact of precommercial thinning on fire behavior, but also identified the rate of decomposition of the generated debris. Through the study, they were able to demonstrate that the removal of canopy fuels limited the propagation of active crown fires, and that after two years the amount of dead surface fuel was reduced to a level that limited fire crowning. Still, they admitted that they did not consider the evolution of the understory, even when shrubby fuels accumulated drastically once the overstory canopy was opened [18].

The objective of this study is to evaluate the impact of implementing early precommercial thinning on Aleppo pine thickets, both on fuel structure and fire behavior over time (immediately after the treatments and 2-, 4-, 6-, 8-, and 10-years post-treatment). For this purpose, we used: (1) measurements on living and dead canopy and surface in all size classes and (2) simulations of fire behavior under different weather conditions to identify the longevity of thinning, pruning, and surface fuel cutting as effective fire prevention measures. We expected to find trade-offs between variables defining the evolution of living and dead fuels, provide a more complete view of precommercial thinning effectiveness, and when possible, suggest potential improvements to this extended practice.

## 2. Materials and Methods

### 2.1. Study Site and Field Measurements

The study area is located in the *Serra de Rubió i Montserrat* (4 1°42′ N, 1°36′ E), Catalonia, Spain. This area was affected by a large forest fire in 1986, and is currently dominated by dense post-fire regeneration of Aleppo pine, with some individual examples of *Quercus ilex* (Holm Oak). The area has a typical Mediterranean climate with mild winters and warm summers, and a seasonal rainfall regime.

During the last decade, large portions of the recovered forest have been subjected to mechanical treatments, with the main aim of forest fire prevention and improving forest vigor and stability (Figure 1). Those treatments consisted of: (1) a precommercial thinning to reduce the tree density to 1000–1200 trees per ha, uniformly distributed; (2) cutting slash to pieces no longer than 1 m, and placing them on the ground, avoiding accumulation over 50 cm in height; (3) pruning trees to 1/3 of the tree height; (4) cutting understory shrubs (seldom implemented, as the initial density of stands limited the development of understory vegetation). Treatments were implemented annually across the study area from 2006 to 2016, allowing the examination of fuels for up to ten years since the treatment.

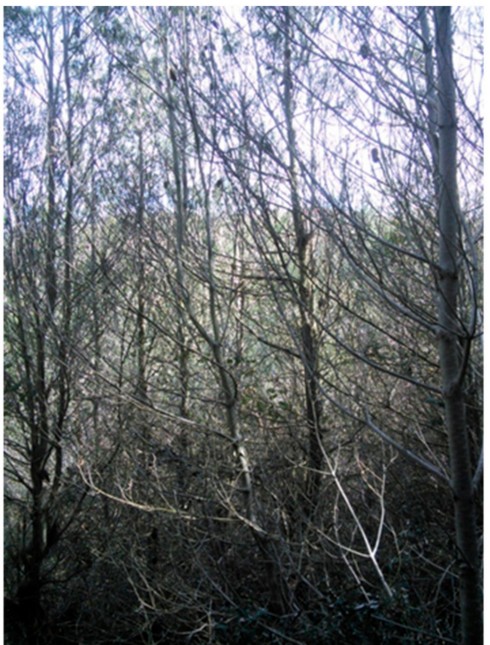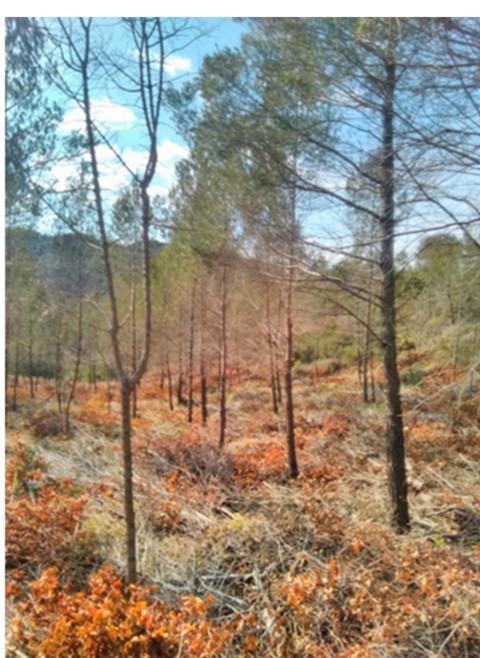

**Figure 1.** Untreated forest (T0) and forest shortly after the mechanical treatments were implemented (T0.5).

During the summer of 2016, a network of 44 plots was established to capture the evolution of the effectiveness of these treatments in terms of fuel hazard and potential fire behavior. The plots were distributed across areas that were treated in different years, so as to capture the evolution of the variables defining forest structure and the amount and distribution of surface fuels. The measured plots were assigned a code according to the time since treatment: Code T0 for the seven untreated plots; T0.5 for the seven plots treated during the previous winter; T2 for six plots treated during summer 2014; T4 for the six plots treated during summer 2012; T6 for the six plots treated during summer 2010; T8 for the six plots treated during summer 2008; and T10 for the six plots treated during summer 2006.

The sampling method consisted of a set of 8-m circular radius plots to measure tree characteristics, including the diameter at breast height (DBH) for all trees in the plot, and tree height (TH), crown base height (CBH), and crown width (CW) for the three trees nearest the plot center. Shrub and slash cover was measured in twenty 50 by 50 cm subplots arranged in a 10 by 0.5 m transect that crossed through the center of each circular plot. Surface fuel loading was measured in five 1 by 1 m subplots by destructively sampling litter, understory, and wood debris, and weighing samples after drying in a laboratory at 80 degrees C for 48 h. Final measurements were classified by fuel type (litter, herbaceous, shrub, and woody debris), with the dead fuels being further divided by time-lag sizes (1 h, ≤0.6 cm; 10 h, 0.6–2.5 cm; 100 h, 2.5–7.6 cm).

Representative indicators were estimated per plot for all of the measured values, as a direct measurement or as a means of observations when based on measurement repetitions within a plot. In addition to the measured values, the canopy bulk density per plot was calculated as the mass of available canopy fuels per unit of canopy volume, with the crown biomass being estimated using allometric functions for Aleppo pine and also for Holm oak, when present as an accompanying species, [19,20]. Differences and similarities between time since treatment groups were tested applying Tukey's HSD test.

### 2.2. Fire Behavior Simulations and Weather Scenarios

Potential fire behavior was estimated using the stand level fire behavior model NEXUS [21]. Nexus uses existing models of surface and crown fire behavior, indicating the potential for crown fire, as well as the expected fire spread and intensity depending on fuel characteristics and weather-related variables [22].

As additional inputs to run NEXUS, observed fuel loads were used to create custom fuel models. Fine woody fuel load was obtained using Casals et al. [18] equations. Fuel bed depth was calculated from understory transect samplings. Surface area to volume ratios, plus dead and live heat content were estimated based on Domenech [23]. Fire weather scenarios (Appendix A) representing the average and severe weather conditions of the summer months were extracted by evaluating historical records (1999–2015) from the closest meteorological station. For this purpose, we extracted the 50th and 90th percentiles of maximum daily temperatures and wind speed, and minimum relative humidity. Wind speeds were corrected afterwards to account for height differences between measurements and NEXUS simulation needs [24], and the impact of tree canopies [22]. From the weather percentiles, 1 h, 10 h and 100 h fuel moisture contents were calculated following Rothermel [25], while live woody fuel and foliage moisture was defined using regional specific data. Slope and wind direction were fixed at 30% and 0°, respectively, to capture only the effect of the treatment and its evolution over time.

### 3. Results

#### 3.1. Forest Structure and Fuel Characteristics

The results on the dynamics of stand characteristics and fuel loads are shown in Table 1. Immediately after the treatment, the tree density and basal area of the thinned stands were reduced, following the principles of the planned treatment. At the same time, the trees' mean diameter and height increased, showing that the thinning targeted the smaller trees, leaving the dominant trees with more growth potential. For those stands where the treatments were implemented, no significant variation in basal area or mean diameter could be identified over time since treatment. This may be explained by the fact that there is often a delay in the increased diameter growth after thinning, or the potential redistribution of biomass pools after management [26]. Trees often first expand their foliar and root system before exhibiting diameter growth [26]. This expansion of branch length and foliar growth can be observed in the values of canopy cover and crown width, which increased with treatment age (Table 1). Canopy base height did not vary between treated and untreated stands. This lack of pruning effect may be explained by self-pruning in the untreated stands. However, Aleppo pine is not known for having an intense self-pruning trait [27]. Finally, CBD was clearly affected by the treatments, in most cases being reduced by 50% compared to untreated plots. Still, CBD did not seem to vary with treatment age, but rather in terms of variations in treatment intensity, or the number of remaining trees.

Regarding the fuel characteristics, it is clearly visible that shortly after the treatments there was a large accumulation of dead fuels of all sizes (Table 1). As expected, and following similar trends as in the case of Palmero-Iniesta et al. [16], the load of those fuels decreased over time after treatment, due to decomposition processes. On the other hand, shrubs really did benefit from the opening of the canopy, and their associated fuel loads and height increased as time elapsed from the tree removals. Similar to the decay of dead fuels, this increase in the amount of shrub fuels was expected and followed the principles of previous studies from the region [18,28].

#### 3.2. Potential Fire Behavior

By applying the NEXUS fire model to the studied stands, it was possible to obtain a set of relevant parameters on potential fire behavior (Figure 2). For a fixed slope and under average and extreme fire weather conditions, it was possible to identify the effectiveness of the treatments in reducing fire hazard. Using any of the weather scenarios (50 and 90 percentiles), it was clearly visible that all of the fire behavior parameters decreased drastically immediately after the treatment. During the following two years, the average value of any of the fire behavior indicators remained at reasonably low levels regardless of the weather conditions. Once the time since treatments reached four years, all of the fire indicators related to fire behavior doubled in magnitude. Those indicators still increased their value in the sixth year after the treatment, afterwards maintaining an almost steady

state. In most indicators (rate of spread, flame length), the impact of the treatment was still positive in taming fire behavior, regardless of the timespan. Even so, over time, some of those indicators almost went back to pre-treatment levels, especially when extreme fire weather scenarios and high winds were applied.

**Table 1.** Fuel characteristics of the stands depending on the time since treatment (mean ± std.dev) The forest structure variables are as follows: number of trees per hectare; mean diameter at breast height (Dbh); basal area (BA); horizontal projection of the canopy coverage (CC); mean tree height (TH); crown base height (CBH); diameter of the tree canopies (CD); canopy bulk density (CBD); and depth of the surface fuels (FBD). The variables related to the fuel load are as follows: the loads of the dead fuels according to their size (Total plus 1, 10, 100, 1000-h timelag); and the loads of life herbaceous fuels (WLh) and woody fuels (WLs).

| | | T0 | T0.5 | T2 | T4 | T6 | T8 | T10 |
|---|---|---|---|---|---|---|---|---|
| *Stand Structure* | | | | | | | | |
| Density | (trees/ha) | 12117 ± 1811 b | 1293 ± 73 a | 1119 ± 58 a | 1097 ± 182 a | 1471 ± 132 a | 1368 ± 98 a | 1401 ± 136 a |
| Dbh | (cm) | 4.9 ± 0.5 b | 10.5 ± 0.4 a | 12.2 ± 0.6 a | 10.5 ± 1.0 a | 10.1 ± 0.6 a | 11.3 ± 0.3 a | 11.9 ± 1.1 a |
| BA | (m$^2$/ha) | 21.0 ± 2.2 b | 11.24 ± 1.05 a | 13.0 ± 1.0 a | 9.1 ± 1.6 a | 11.6 ± 1.1 a | 13.7 ± 1.4 a | 15.0 ± 1.5 ab |
| CC | (%) | 92 ± 2 b | 68 ± 6 a | 70 ± 3 a | 70 ± 3 a | 72 ± 6 a | 78 ± 4 ab | 81 ± 2 ab |
| TH | (m) | 7.8 ± 0.4 a | 8.7 ± 0.4 a | 9.4 ± 0.5 a | 8.4 ± 0.4 a | 9.0 ± 0.4 a | 9.1 ± 0.4 a | 8.7 ± 0.5 a |
| CBH | (m) | 3.5 ± 0.6 a | 3.4 ± 0.4 a | 3.9 ± 0.7 a | 2.6 ± 0.2 a | 3.0 ± 0.2 a | 2.8 ± 0.3 a | 2.7 ± 0.3 a |
| CD | (m) | 2.1 ± 0.4 b | 3.6 ± 0.4 a | 3.4 ± 0.2 a | 3.9 ± 0.2 a | 3.9 ± 0.4 a | 4.1 ± 0.2 ab | 3.3 ± 0.1 ab |
| CBD | (kg /m$^3$) | 0.16 ± 0.03 b | 0.06 ± 0.01 a | 0.09 ± 0.03 a | 0.05 ± 0.01 a | 0.08 ± 0.01 a | 0.07 ± 0.01 a | 0.05 ± 0.01 a |
| FBD | (cm) | 100.7 ± 12.2 abc | 64.3 ± 8.6 ab | 55.3 ± 15.1 a | 96.4 ± 12.7 abc | 97.8 ± 9.5 abc | 121.1 ± 4.8 c | 109.0 ± 10.2 bc |
| *Fuel Load* | | | | | | | | |
| WT | (t/ha) | 40.1 ± 1.1 a | 107.6 ± 12.7 c | 66.2 ± 8.8 ab | 59.1 ± 9.0 ab | 78.2 ± 11.7 bc | 55.5 ± 4.6 ab | 56.2 ± 7.2 ab |
| W1h | (t/ha) | 33.8 ± 1.6 a | 50.8 ± 3.4 b | 37.8 ± 3.8 ab | 31.2 ± 4.4 a | 38.8 ± 3.8 ab | 28.2 ± 3.9 a | 27.2 ± 4.9 a |
| W10h | (t/ha) | 3.0 ± 1.1 a | 22.68 ± 3.26 d | 14.7 ± 3.3 cd | 10.1 ± 2.7 bcd | 14.2 ± 2.7 d | 4.3 ± 0.8 ab | 6.1 ± 1.1 ac |
| W100h | (t/ha) | 0.2 ± 0.2 a | 24.7 ± 11.9 a | 11.1 ± 4.2 a | 6.3 ± 2.5 a | 14.8 ± 4.0 a | 9.8 ± 3.5 a | 5.6 ± 3.7 a |
| WLh | (t/ha) | 0.4 ± 0.2 a | 0.3 ± 0.1 a | 0.3 ± 0.1 a | 0.5 ± 0.1 a | 0.3 ± 0.1 a | 0.3 ± 0.1 a | 0.6 ± 0.2 a |
| WLs | (t/ha) | 2.8 ± 0.8 b | 0.1 ± 0.1 a | 2.3 ± 2.0 a | 8.4 ± 1.5 bc | 10.1 ± 3.3 bc | 13.0 ± 2.6 c | 14.0 ± 3.1 c |

Means followed by the same letter in a row are not significally different according to the HSD Tukey test.

A clear example of the effectiveness of the treatments was observed when identifying the impact of weather scenarios on the expected fire type (surface fire, passive crown fire, active crown fire). From the simulations, it was clear that the treatments always reduced the possibility of fires spreading actively through the canopies of trees, and only when wind velocity exceeded 40 or 50 km/h did the impact of treatments become less effective (Figure 3). During the first two years, the treatments had a major impact on reducing the intensity of fires, and reducing the risk of fire crowning and spreading through canopies under the most common weather conditions in the fire season (50 percentile, and winds under 30 km/h). It is probable that this positive impact on fire behavior continued until year three after treatment (although this was not studied), as year four can be identified as a transition year between the first years after treatment, when the preventive measures had a clear and recognizable impact on fire behavior, and the remaining ones (years 6, 8, 10) where the stand dynamics minimized the efficiency of the implemented management, regarding fire prevention.

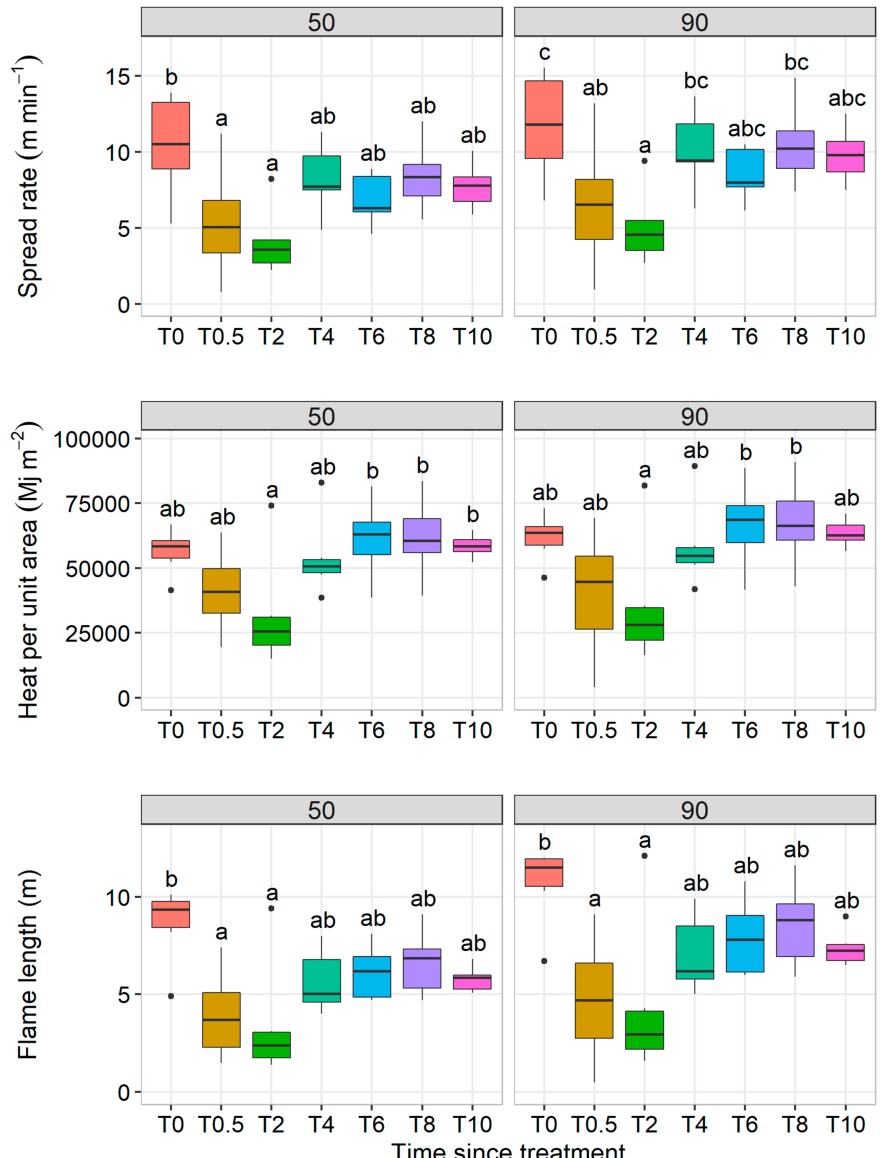

**Figure 2.** Fire behavior parameters for the stands depending on the time since treatment and weather scenarios. Means with same letter are not significantly different according to the HSD Tukey test.

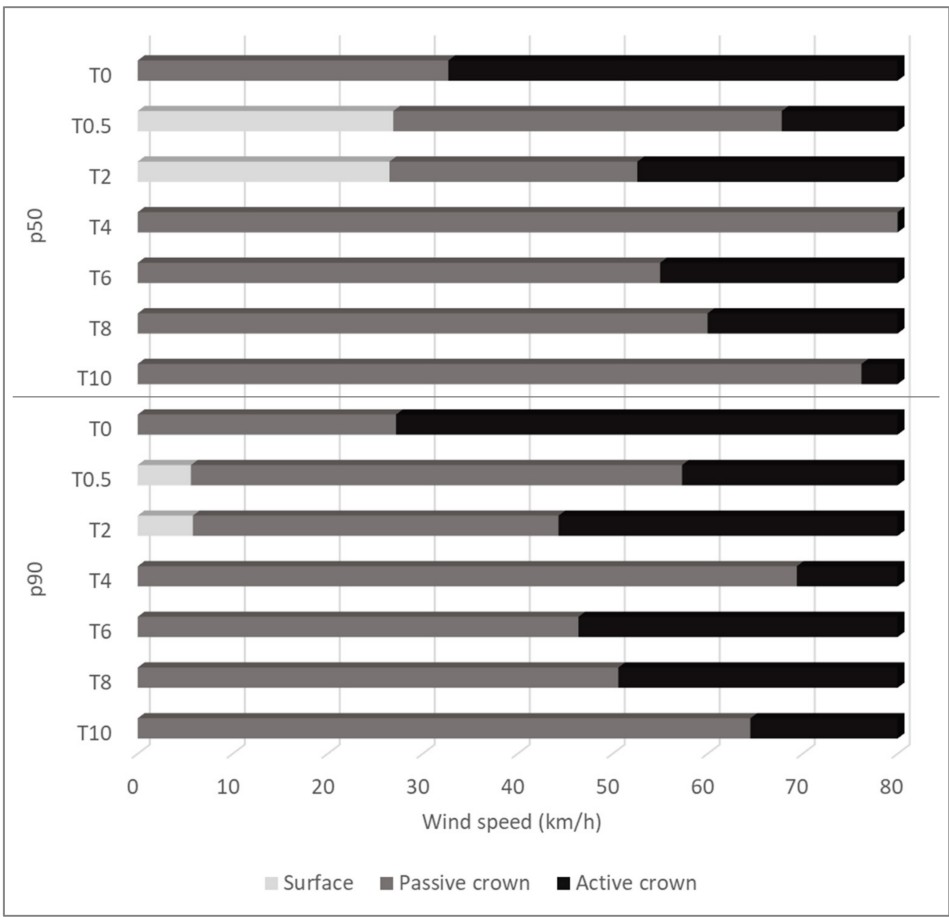

**Figure 3.** Expected fire type depending on the time since treatment, weather scenarios, and wind speed.

## 4. Discussion

　　This study evaluates the temporal effectiveness of precommercial thinning on dense *Pinus halepensis* stands. As expected, thinning combined with pruning decreased the potential of sustaining active crown fires, due to the reduction of crown biomass and horizontal crown continuity [29,30]. Although this positive impact on crown fuel modification was sustained over time, other positive aspects associated with thinning, regarding taming fire behavior, were not that clear as the time since treatment increased. During the first two years following treatment, the thinning, pruning and understory cutting resulted not only in the mentioned changes in the canopy fuels, but also in a reduction of the surface fuel depth. By generating an additional distance between the surface fuels and the canopies, the probability of fire crowning declined significantly [22]. This relevant reduction in fire hazard was even further enhanced by the decomposition of dead fuels over time [16,31], which obviously occurred immediately after the thinning [32,33]. As prognosticated by Palmero-Iniesta et al. [16], an additional measurement of the understory development was required to evaluate the effectiveness of this type of treatments over time. Opening the canopy resulted in the growth of shrubs [18,28], and by the fourth year the vertical gap between surface and crown fuels returned to pre-treatment stages. Subsequently, most of the positive impact on fire behavior that Palmero-Iniesta et al. [16] showed in their study, might be lost over time once the evolution of living surface fuels is added to the equation (Figure 4).

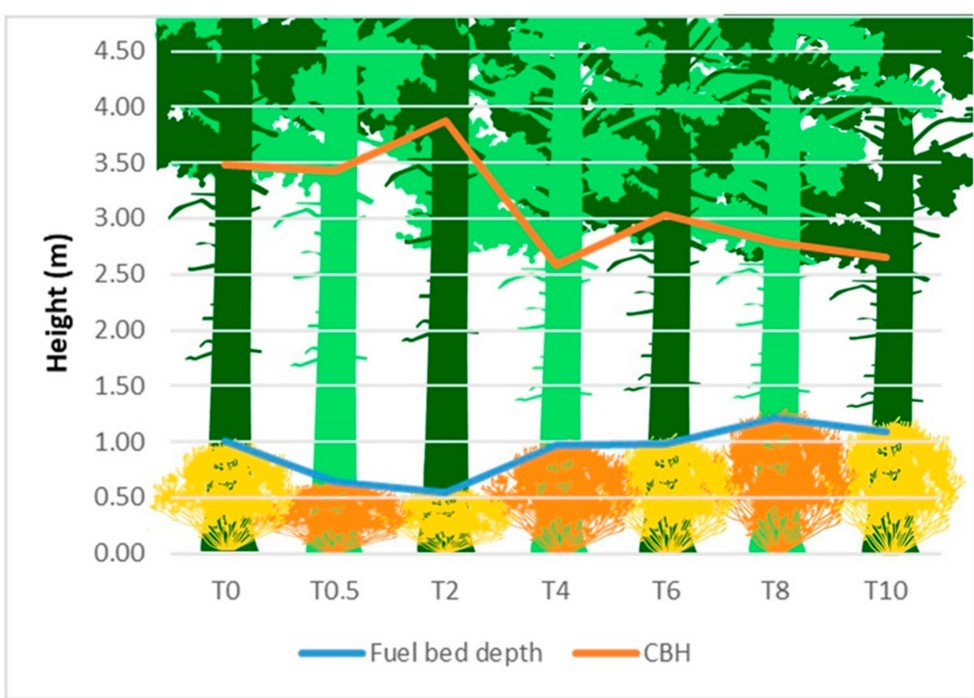

**Figure 4.** Evolution of the vertical fuel gap between surface and canopy fuels, which can be considered one of the main factors defining the possibility of fire crowning.

Precommercial thinning is a valid method to reduce canopy fuel loads and their horizontal continuity. It is also clear that once a fire reaches the canopies, under extreme conditions of foliar moisture and wind speed, it is difficult to ensure that the treatment will accomplish its main objective of preventing crown fire propagation. When considering this type of treatment as a preventive tool to increase the resistance of a forest landscape to fire, we must plan its implementation within a temporal and spatial context. The initial objective of an early precommercial thinning should be to improve the stand characteristics in the future by preventing stagnation and improving the growth and vitality of remaining trees. Early thinning should favor the growth and reproductive capability of the uncut trees [34,35]. Modifying the spacing of trees through thinning also improves the stability of the forest [36,37] and reduces the stress often caused to this species by drought [17,38]. The expected improvement in the future vigor of the forest can minimize the risk of cascading effects due to iterated disturbances [39] by modifying the disturbance recurrence patterns and the trees' resistance when those disturbances occur [40]. With a more immediate and fire-orientated point of view, it is clear that the evolution of bushes after thinning poses a threat in terms of fire crowning, and periodically removing those fuels through prescribed burning [18] or through the use of cattle [41] must be taken into account if young thinned stands are to be considered fire resistant. Contemplating all of these aspects related to the effectiveness of early thinning in mitigating fire impact, together with some observations from our study, it should be considered that although the effect of the currently proposed treatments is in general positive, further research may be required to improve silvicultural recommendations in this regard. For example, implementing the first thinning at more advanced stages of development may help to identify clear dominance patterns between trees and facilitate self-pruning, thus improving the efficiency of the practice.

Even if thinning, especially when accompanied with bush control operations, has proven its value to reduce fire hazard, no one treatment can guarantee protecting an Aleppo pine stand from fire. In order to disrupt intense fire propagation through a forested landscape, especially one dominated by a highly flammable tree species [42], it is necessary to create a level of spatial heterogeneity in forest treatments and for other land uses. By implementing treatments strategically across a landscape, it is possible to have an impact

on fire propagation [43,44]. Treatment patches may consider changes in land use, as the most drastic and effective type of fuel modification, however, changing the species can also operate as way to mitigate extreme fire behavior. Intercalating stands dominated by different tree species, providing the traits are appropriate, has shown its potential to decrease fire intensity and severity [45]. Creating mixtures of species within the stand, also has been mentioned as a fire prevention measure [46], but the impact of this practice can be limited if the most flammable species has a dominant role in defining flammability within the stand [47].

## 5. Conclusions

When assessing the effectiveness of early precommercial thinning to thickets on mitigating fire behavior, the effect that opening the canopy has on bush development cannot be underestimated. Early thinning has a narrow window of effectiveness regarding fire control. Short-term mitigation objectives should not be the focus when planning this type of treatment. Instead, the emphasis should be on the potential long-term evolution of the forest and how it fits into a full silvicultural itinerary, to achieve a more complex set of objectives related to the yield of different ecosystem services.

Managing a large forest, resulting from post-fire regeneration, should not depend on preventive silviculture alone. The effective fire smart management of forested lands should include different scale strategies, ranging from the stand composition and structure to the arrangement of the forest and other land-uses across a landscape, including the initial post-fire strategies aimed at controlling the impact of tree cover loss and the long-term strategies aimed at obtaining a desired landscape that is resilient and a provider of multiple functions.

**Author Contributions:** Conceptualization and field work, M.P. and E.B.; analysis and first draft sketching M.P. and E.B.; writing and editing of final draft J.R.G.-O.; review all. All authors have read and agreed to the published version of the manuscript.

**Funding:** This research has received funding from the European Union's Horizon 2020 Research and Innovation Programme (grant agreement number 101037419—FIRE-RES project).

**Data Availability Statement:** Not applicable.

**Acknowledgments:** Special thanks to Diputación de Barcelona (DIBA) and Life Montserrat, Rut Domenech and Mario Beltrán from the Forest Science and Technological Centre of Catalonia (CTFC) and Asier Larrañaga from GRAF fire fighters of the Catalan Government. Also, to ForRes project (RTI2018 098778-BI00) from MICINN, and to the Agency for the Research Centre of Catalonia (CERCA Programme/Generalitat de Catalunya). Field data collection was completed with the assistance of Sonia Navarro and Moisés Rial.

**Conflicts of Interest:** The authors declare no conflict of interest.

## Appendix A

**Table A1.** Weather and fuel moisture conditions used in fire behavior simulations, related to 50th and 90th percentiles of temperature, relative humidity and wind speed.

| Variable | 50th Percentile | 90th Percentile |
|---|---|---|
| 1 h Fuel Moisture (%) | 8 | 6 |
| 10 h Fuel Moisture (%) | 9 | 7 |
| 100 h Fuel Moisture (%) | 10 | 8 |
| Live Woody Fuel Moisture (%) | 95 | 75 |
| Foliar moisture (%) | 120 | 105 |
| Mean wind speed (km/h) | 6.9 | 9.3 |
| Maximum wind speed (km/h) | 30.0 | 38.5 |
| Temperature (°C) | 31.1 | 35.0 |
| Relative humidity (%) | 32 | 20 |
| Days with these conditions [1] (%) | 52.5 | 10.7 |

[1] related to summer months (June, July, August).

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
