# Peer review of "Dynamic Evaluation of Early Silvicultural Treatments for Wildfire Prevention"

_forests, doi:10.3390/f13060858_

Round 1
Reviewer 1 Report
The MS reports on a study to assess fuel and canopy characteristics and possible effects on fire behavior in stands of different ages since mechanical thinning treatments. Overall it is a very interesting study with somewhat unexpected results especially in the older stands.
However, I have two major concerns. First and the most major is that some sort of statistical tests of the results of the varying time-since-thinning plots must be conducted for a reader to truly trust differences argued for the specific characteristics measured and modeling results. This request is for Tables 1 and 2 and figure 2. For example, a very simple approach I suggest would be an ANOVA with post-hoc pairwise comparisons using a Tukey honest significant difference. Also it would much more useful for a reader to see table 2 especially as a figure (and perhaps table 1 as well; both of the tables could be moved to the end of the paper as an appendix to have the actual numbers but a visual presentation is much easier to see the trends). I suggest to present these data as box-and-whiskers plots; these are easy to do in Excel (instructions available online). However, no matter presentation in tables or figures some sort of statistical test of data like these has to be done to determine if what our eyes tell us is actually correct interpretation of the data.
Second, the MS needs to have both a much better spell check and it should be edited by a native English speaker. For examples: Abstract: “over time” (here and elsewhere) is two words, not one. Pinus halepensis (and other binomial names) must to be italicized (here and elsewhere). Also add in parentheses that Aleppo pine is the common name if referred to this way in the rest of the paper (and “Aleppo” is the commonly accepted international spelling). “established” not “stablished”. L35: “…impact of silvicultural actions is their”… After this I have not tried to correct the English writing.
Other comments:
Table 1: DBH is mean DBH of measured trees? Better would be to use a metric such as quadratic mean diameter.
Results section: There are numerous instances in the results that should be moved into the discussion. “Results” should report “just the facts”; discussion is where interpretations of those facts must reside. For examples, L163: “This lack of impact on tree size can be explained…”; L172: “This lack of impact of the pruning could be explained by…”. These and following sentences must be moved to the discussion.
Fig 2: It would be easier to see trends in these results if they were grouped by weather conditions, not time-since-thinning; group all the 50% results together, then the 90%.
Fig 3: A very effective figure, nicely presented, but again I wonder if these trends are significant and I’m not exactly clear on why there is a decrease in CBH; I think the explanation given is split too much between what was mentioned in the results vs what is in the discussion; moving all the interpretations from the results to the discussion may help with my confusion here.
Author Response
Answer to reviewers.
We thank the reviewers for their highly constructive comments, that in our opinion helped to improve the overall quality of the manuscript.
Following the recommendations from both reviewers, we included statistical tests to the study, and sent the manuscript for language revision.
Regarding specific comments from Reviewer 1:
A general correction of English has been done by a native English speaker. Also, we added statistical tests (Table 1 and Table 2), converted results from Table 2 to a figure, and modified figure 2 (new figure 3) as recommended.
We also modified through all the paper the use of P. halepensis – Aleppo pine as recommended, and explained in Table 1 that the variable DBH refers to mean diameter.
The only aspect we did not consider from Reviewer 1 was the comment: Results section: There are numerous instances in the results that should be moved into the discussion. “Results” should report “just the facts”; discussion is where interpretations of those facts must reside. For examples, L163: “This lack of impact on tree size can be explained…”; L172: “This lack of impact of the pruning could be explained by…”. These and following sentences must be moved to the discussion.
In this case we did maintain the sentence as recommended by Reviewer 2. Although we understand there is certain interpretation of results, and may fit the discussion part, we consider that our discussion was focused on more general aspects of the management impact, without focusing on specific variables. Move this paragraph to discussion would require extensive modifications on the manuscript style, and we hope that modifying the manuscript as recommended by Reviewer 2 will be enough.
Reviewer 2 Report
The paper represents an important contribution by studying the longevity of fuel treatments in a Mediterranean forest. However, there are several areas for improvement before the piece can be considered for publication.
- The authors should consult with a statistician and include some statistical analysis before they comment on differences between treatments.
- Throughout the paper the authors refer to 'early thinning', but this term is never defined.
- Extensive edits to the language are needed and I attempt to make suggestions below, but it would be good to have another native English speaker read the draft. Chose one tense throughout, preferably past tense.
- Throughout, the authors refer to 'before' and 'after' treatment, which implies a time series instead of a space for time study. Instead, they should refer to 'untreated' and 'treated areas.
Line 12: Italicize Pinus halepensis here and throughout and use the full name: Pinus halepensis Mill. You can put Aleppo pine in parentheses after, and then use Aleppo pine throughout the manuscript.
Line 15 - 16: Suggested language - " The plots were at different stages of fuel evolution (0.5 to 10 years since treatment)"
Lines 21-22: Suggested language - "After hat initial period, the buildup of understory vegetation minimized treatment effectiveness in mitigating potential fire behavior"
Line 35: replace 'it's' with 'is'
Line 39: replace 'ground' with 'surface'
Line 43: delete 'apply' and 'early' (unless you define early thinning), change 'to dense young stands' to 'of dense young stands'
Line 45: Suggested language - 'between neighboring trees and for active crown fire'
Line 47: change ' translate on thicker bark' to 'translate to thicker bark'
Line 57: 'in Catalonia'
Line 58 - 59: suggested language - 'the tendency for high intensity fire resulting in high rates of tree mortality'
Line 61 -64 - suggested language ' Of the management strategies considered to diminish the probability of hazardous crown fires, thinning of highly dense young stands is one the of most commonly used, due to the combined..."
Line 66 - add 'over time' after 'type of practices and delete 'following this idea'
Line 69 - delete 'not only the'
Line 70 - delete 'also' and replace 2 with 'two', place a comma after two, and insert 'surface' before dead fuel.
Line 71 - delete ' was prone to initiate them' and replace with promoted ignition (if in fact the study looked at potential for fire initiation, and not some other fire behavior metric)
Line 72-73: suggested language - 'even when shrubby fuels accumulated drastically after opening the overstory canopy'
Line 75-76: delete 'at short and mid-term' and replace with 'over time'
Line 77-83: suggested language - ' measurements of living and dead canopy and surface fuels of all size classes and 2) simulations of fire behavior under different weather conditions, to identify the longevity of thinning, pruning, and surface fuel cutting as effective fire prevention measures'
Line 86: insert 'in' before Montserrat
Line 87: "This area was affected by a large forest fire in 1986, which burned nearly 86% of the landscape". Not sure if 'landscape' is the right term here, since I'm not clear on what you are trying to say. 86% of the study area?
Line 88: Replace 'nowadays' with 'currently'
Line 89: Italicize Quercus ilex and provide full name.
Line 93: Replace 'on' with 'of'
Line 95: Replace 'place' with 'placing' and add comma after ground
Line 96: Cutting understory shrubs?
Line 98-99: "Prior to treatment, the forests consisted of dense even-aged stands (over 10,000 tress per ha), regenerating shortly after the 1986 fire" - this sentence should be moved to the previous paragraph.
Line 99-102: "Treatments were implemented annually across the study area from 2006 to 2016 (?), allowing for an examination of the accumulation of fuels for up to ten years since treatment." - I would also move this earlier in the methods section.
Line 104: "evolution of treatment effectiveness in terms of fuel hazard and potential fire behavior'
Lines 115-123: suggested language ' The sampling method consisted of a set of 8 meter circular radius plots to measure tree characteristics, including diameter at breast height (DBH) for all trees in the plot and tree height (TH), crown base height (CBH) and crown width (CW for the three trees nearest to plot center. Shrub and slash cover was measured in 40(?) 50 by 50 cm subplots arranged in a 10 by 0.5 m transect that crossed through the center of each circular plot. Surface fuel loading was measured in 5 1 by 1 m subplots by destructively sampling litter, understory vegetation, and woody debris and weighing samples after drying in a laboratory at 80 degrees C for 48 hours. Final measurements were classified by fuel type (litter, herbaceous, shrub, woody debris) and dead fuel size class (1 hr....
Line 131: delete 'each'
Line 147: delete 'lower' and 'higher', replace 'humidifies' with 'humidity'
Line 148: replace 'being' with 'were', delete 'afterwards'
Line 149: Not sure what you mean by correcting for differences between measured and simulated wind heights. You'll need to explain this.
Line 156-157: change to Table 1, insert 'tree' before density
Line 159-160: delete 'as usually recommended'
Line 161: delete 'alive', replace 'have been' with 'were'
Line 162: replace 'on basal area' with 'in basal area'; replace 'depending on the' with 'with'
Line 163-179: suggested language ' This may be explained by the fact that stands all initiated at the same time, a delay in the increased diameter growth commonly seen after thinning, or the potential redistribution of biomass pools after management (26). Trees often first expand their foliar and root system before exhibiting diameter growth (26). This expansion of branch length and foliar growth can be observed in the values of canopy cover and crown width, which increased with treatment age (table 1). Canopy base height did not vary between treated and untreated stands. This lack of pruning effect may be explained by self-prucing in the untreate stands. Although Aleppo pine is not known for having intense self-pruning as one of its traits (27). Finally, CBD was clearly affected by the treatments, being in most cases reduced by 50% compared to untreated plots. Still, CBD did not seem to vary with treatment age, but rather on variations in treatment intensity, or the number of remaining trees'
Line 184-186: 'Herbaceous fuel loads do not follow any clear temporal trend' - May want to note that this is likely influenced by annual climatic conditions.
Line 188 - delete ' in a lesser extend their'
Line 205 - replace 'after the' with 'since'
Line 206-208: Replace 'their' with 'in'. Delete ' those indicators still increased their value on the sixth year after treatment, keeping afterwards an almost steady state' - This kind of interpretation is inappropriate without any statistical analyses.
Line 210: pre-treatment has an underscore instead of a slash. Replace 'stages' with 'levels'
Line 235: replace 'to' with 'with'
Line 237-238 suggested language - thinning, pruning, and understory cutting
Line 242: What do you mean by fuel compacting? How was this accomplished?
Line 248: replace 'return to pre-treatment stages' with 'was similar to untreated areas'
Line 271 - 273: what do you mean by improving the stand evolution and characteristics on the future and stability of the forest? Need to be more specific on treatment objectives.
Line 276: change 'cascade' to 'cascading'
Line 279-280 - Not sure what you mean by implementing more intense thinning later? This idea isn't developed in the paper and I'm not sure why you would cite Agee here. You need to be more specific on the strategy you are trying to describe and cite the relevant literature.
Line 288: replace 'on' with 'in'
Line 293-302: suggested language '...fire hazard, any one treatment cannot assure an Aleppo pine stand can be protected from fire. In order to ...it is necessary to create a level of spatial heterogeneity in forest treatments and other land uses. By implementing treatments strategically across a landscape, it is possible to have an impact of fire propagation. Treatment patches may consider land use changes..... but also shifts in species can operate as mitigators of extreme fire behavior. Mixed stands ..."
Line 305: replace ' on defiing stand flammability' with 'in the stand.
Line 312 'it fits' instead of 'they fit'; replace 'itinery' with 'strategy'
Line 314-316 - this idea of delaying thinning is not well articulated in the paper or backed up with literature. Expand upon the idea or delete.
Line 321 - delete this
Line 322 - 'from' not 'form', 'stand' not 'stands'
Author Response
Answer to reviewers.
We thank the reviewers for their highly constructive comments, that in our opinion helped to improve the overall quality of the manuscript.
Following the recommendations from both reviewers, we included statistical tests to the study, and sent the manuscript for language revision.
Regarding specific comments from Reviewer 2:
We thank Reviewer 2 for the intensive work on suggesting style and language corrections. We included all the recommendations from reviewer 2 before send the manuscript for professional language revision.
Regarding general comments:
- We have included statistical analysis in Table 1 and Table 2 (in the revised version changed to Figure 2)
- We have completed the definition of early thinning by using the term of pre-commercial thinning to thickets, a more common and known term.
- We have revised the consistence on the use of Aleppo pine term.
- We have included the terms untreated and treated
As mentioned, all corrections and suggestions were considered, and in almost all cases included exactly as recommended.
The only suggestions we did not included as Reviewer 2 suggested are:
- Line 86: insert 'in' before Montserrat. (NOW L.82)
We maintained the name but set it into italics as is the name of the area in Catalan language.
- Line 184-186: 'Herbaceous fuel loads do not follow any clear temporal trend' - May want to note that this is likely influenced by annual climatic conditions.
We removed the line. As the reviewer suggested, management has not much impact on herbaceous fuel (at least considering the study temporal frame), but annual climatic conditions, so we think that showing or explain the evolution of those fuels overtime does not add relevant information.
- Line 279-280: not sure what you mean by implementing more intense thinning later? This idea isn't developed in the paper and I'm not sure why you would cite Agee here. You need to be more specific on the strategy you are trying to describe and cite the relevant literature.
We agree that the idea is not justified with our design or results, and is not relevant for the study. Instead of elaborating or explaining this concept, we decided to remove the line.
- Line 314-316: this idea of delaying thinning is not well articulated in the paper or backed up with literature. Expand upon the idea or delete.
We also agree that the idea is new and not justified with our results or the design of the experiment. We deleted the line in the conclusions, as suggested.
Round 2
Reviewer 2 Report
The manuscript is much improved and the authors have adequately addressed the concerns of both reviewers. However, there are still some minor edits the authors need to consider.
Title – change ‘early’ to ‘precommercial’
Abstract – line 11; abiotic and biotic disturbances
Abstract – line 13; put ‘Aleppo pine’ in parentheses after Pinus Halepensis
Abstract – line 22; replace ‘early thinning’ with ‘precommercial thinning’
Line 83 – should read ‘strategy consists of…”
Lines 91, 96, 110, 121, 876, 995, 1202 – change ‘early thinning; to ‘precommercial thinning’
Line 111 – ‘extend the effectiveness of this practice’
Line 117 – ‘the area has a typical Mediterranean climte…’
Line 347 – delete ‘the’ before treatment
Line 370 – replace ‘Queercus ilex’ with ‘Holm oak’
Line 445 – At the end of the methods, you need to include a description the statistical tests you used in your analysis.
Table 1 description – what are the numbers describing? Mean and standard error?
Line 727 – change ‘fire intensity’ to ‘fire behavior’
Line 1189 – add ‘uses’ after land
Line 1209 – ‘include strategies at different spatial and temporal scales’; replace ‘stands’ with ‘stand’
Author Response
We thank Reviewer 2 for this second revision.
All corrections and suggestions have been considered exactly as recommended.
The only suggestion we have not included is:
- Title – change ‘early’ to ‘precommercial’
We have changed all over the paper early thinning by precommercial thinning, but in the case of the tittle “Dynamic evaluation of early silvicultural treatments for wild-fire prevention”, we consider better to keep “early”, because the title refers to silvicultural treatments. Then, in the abstract and paper we explain that the silvicultural treatment implemented in early stages of forest development is precommercial thinning.